# Structures of ZYG11B-EloB-EloC-substrate complex reveal mechanisms of CRL2^ZYG11B assembly and function

Ni Lin[1,2,4], Han Feng[2,4], Yushan Geng[2,3], Yina Gao[2], Miao Shi[1,2], Songqing Liu[2], Pu Gao [2,3] ✉ & Yong Wang [2] ✉

ZYG11B is a substrate receptor of the Cullin2-RING E3 ligase (CRL2), mediating the Gly/N-degron pathway and contributing to diverse processes including cell cycle control, protein homeostasis, apoptosis, and innate immunity. While previous studies resolved the structure of its truncated ARM domain, how full-length ZYG11B coordinates substrate engagement and CRL2^ZYG11B assembly remains unclear. Here, we present cryo-EM structures of full-length human ZYG11B in complex with the EloB–EloC adaptor and a Gly/N-degron peptide, revealing a seahorse-like architecture with distinct interfaces for adaptor and substrate binding. Unexpectedly, ZYG11B adopts both monomeric and dimeric assemblies, with the dimer stabilizing two substrate-binding sites in opposite orientations. Functional assays demonstrate that interfaces mediating adaptor recruitment, substrate binding, and dimerization are essential for substrate degradation, suggesting a dynamic mechanism involving both assembly states. These findings provide a structural framework for understanding CRL2^ZYG11B-mediated ubiquitination and offer mechanistic insights that may inform the rational design of ZYG11B-based applications.

The ubiquitin-proteasome system (UPS), together with autophagy, constitutes the two major cellular protein quality control pathways[1-4]. UPS-mediated degradation is initiated by a hierarchical enzymatic cascade involving E1 (activation), E2 (conjugation), and E3 (ligation) enzymes, which catalyze the ubiquitination of target proteins destined for degradation by the 26S proteasome[5-8]. Substrate specificity is conferred by E3 ligases, which function as either single polypeptides or multi-subunit complexes. Typically, E3s comprise a substrate-binding module that recognizes specific degradation motifs (degrons) and a catalytic module that transfers ubiquitin from E2 to substrates[9-11].

Various types of degrons have been identified, including N-degrons (the well-known N-end rule pathway), internal degrons, and C-degrons, often exposed upon enzymatic cleavage or posttranslational modification[9,12,13]. To accommodate this diversity, E3s have evolved distinct recognition mechanisms. Based on structural and catalytic

features, E3s are broadly categorized into RING (Really Interesting New Gene), HECT (Homologous to E6AP C-terminus), and RBR (RING-between-RING) families. While RING E3s catalyze direct ubiquitin transfer from E2 to substrate, HECT and RBR E3s form a transient E3-ubiquitin thioester intermediate prior to ligation[10,11,14]. Among E3s, Cullin-RING ligases (CRLs) represent the largest subfamily. CRLs are assembled on one of eight Cullin scaffolds (Cul1-3, Cul4A/B, Cul5, Cul7, Cul9), and typically consist of a RING protein (Rbx1 or Rbx2), an adaptor, and a substrate receptor[5,14-19]. This modular architecture enables CRLs to target a broad spectrum of substrates, accounting for up to 20% of UPS-mediated proteolysis[14-16].

Recent study identified ZYG11B and ZER1—two members of the ZYG family—as substrate receptors that recognize the Gly/N-degron, a previously uncharacterized degron type defined by an N-terminal glycine residue, and recruit Cul2 to form the CRL2^ZYG11B and CRL2^ZER1 E3

[1]Science and Technology Innovation Center, Shandong First Medical University & Shandong Academy of Medical Sciences, Jinan, China. [2]National Laboratory of Biomacromolecules, CAS Center for Excellence in Biomacromolecules, Institute of Biophysics, Chinese Academy of Sciences, Beijing, China. [3]University of Chinese Academy of Sciences, Beijing, China. [4]These authors contributed equally: Ni Lin, Han Feng. ✉e-mail: gaopu@ibp.ac.cn; wangyong@ibp.ac.cn

ligases, respectively[20]. In metazoan cells, Gly/N-degrons are typically underrepresented in the proteome and become exposed upon caspase cleavage or defective N-myristoylation, suggesting critical roles for ZYG11B and ZER1 in apoptotic signaling and membrane targeting[20,21]. In addition, the ZYG11B-mediated Gly/N-degron pathway could be utilized during viral infections[22–25]. For instance, the NLRP1 inflammasome is activated when ZYG11B recognizes a Gly/N-degron signal of NLRP1 exposed by enteroviral 3C protease cleavage[22], while SARS-CoV-2 ORF10 protein binds ZYG11B in a Gly/N-degron-dependent manner to influence viral infection through an inclusive mechanism[23–25]. Notably, ZYG11B can also promote degradation of substrates lacking canonical degrons, such as the cell cycle regulator cyclin B1, suggesting additional layers of substrate recognition[26].

ZYG11B contains three domains: an N-terminal von Hippel-Lindau (VHL) box, a central Leucine- Rich Repeat (LRR) domain, and a C-terminal Armadillo Repeat (ARM) domain. Despite previous structural determination of a truncated ARM domain covalently fused to Gly/N degron peptides[27–29], how full-length ZYG11B coordinates substrate binding and adaptor proteins to assemble into the functional CRL2$^{ZYG11B}$ E3 ligase remains unclear.

Here, we present cryo-electron microscopy (cryo-EM) structures of full-length human ZYG11B in complex with a substrate peptide and the EloB-EloC adaptor complex, and further model the intact CRL2$^{ZYG11B}$ holoenzyme. Complemented by biochemical experiments and cell-based assays, these findings demonstrate that ZYG11B mediates both monomeric and dimeric assemblies of the CRL2$^{ZYG11B}$ complex, with dimerization being crucial for efficient substrate ubiquitination and degradation.

## Results

### ZYG11B mediates both monomeric and dimeric assemblies

To elucidate how ZYG11B recruits EloB-EloC adaptor complex and engages substrates, we sought to determine the cryo-EM structures of ZYG11B-EloB-EloC complex bound to the degron peptide, both in the absence and presence of the Cul2-Rbx1 module. To this end, full-length ZYG11B was co-expressed and purified together with EloB and EloC to form the ZYG11B-EloB-EloC core complex, which was subsequently incubated with Cul2-Rbx1 to form the ZYG11B-EloB-EloC-Cul2-Rbx1 holoenzyme complex. Both samples exhibit favorable biochemical behavior and good purity, as assessed by size-exclusion chromatography and SDS-PAGE (Supplementary Fig. 1a). Unexpectedly, two-dimensional classification of the cryo-EM data reveals that both complexes adopt not only monomeric but also dimeric assemblies (Supplementary Fig.1b). Analytical ultracentrifugation (AUC) assays further confirm the coexistence of monomeric and dimeric species in solution, regardless of the expression system or buffer ionic strength (Supplementary Fig.1c), suggesting that both assembly states are intrinsic properties of ZYG11B-containing complexes. Given that some CRL substrate receptors are known to mediate oligomerization[30–35], we speculated that the dimerization process was mediated by ZYG11B itself. In support of this hypothesis, co-immunoprecipitation assays using differentially tagged ZYG11B constructs demonstrate self-association in cells (Supplementary Fig.1d), consistent with the previous report[36]. These findings indicate that ZYG11B can form homodimers both in vitro and in cells.

We next reconstructed near-atomic cryo-EM structures of the monomeric and dimeric ZYG11B-EloB-EloC core complexes bound to a Gly/N-degron peptide (GYIND) derived from the SARS-CoV-2 ORF10 protein, at global resolutions of 3.37 Å and 3.27 Å, respectively (Fig. 1, Supplementary Fig. 2,3 and Table 1). The monomeric complex contains one copy of ZYG11B-EloB-EloC-substrate at a 1:1:1:1 stoichiometry (Fig. 1a, b), while the dimeric assembly is composed of two such units arranged symmetrically (Fig. 1c, d). These structures clearly show that ZYG11B mediates dimer formation, in line with the biochemical and cellular data. Structural alignments indicate that the core module

adopts a similar conformation in the monomer and in each unit of the dimer (Supplementary Fig.4a). Structure of ZYG11B also superposes well with its AlphaFold model (Cα RMSD = 1.56 Å), indicating the reliability of AlphaFold predictions for other ZYG11 family members (Supplementary Fig.4b–d).

### Structure of full-length ZYG11B

The full-length ZYG11B folds into an overall seahorse-shaped architecture, with the N-terminal VHL box and the central LRR domain forming the head region, and the C-terminal ARM domain constituting the body and tail regions (Fig. 2a, b). The LRR and ARM domains pack tightly with each other through intramolecular hydrogen bonding and hydrophobic interactions, thereby stabilizing the overall structure (Fig. 2c). The VHL box contains a canonical BC box that spans the first N terminal helix of ZYG11B, followed by a Cullin box (Fig. 2d). Notably, ZYG11B features an unusually long and extended loop connecting these two motifs, which distinguishes it from other VHL box-containing proteins (Fig. 2d and Supplementary Fig. 5a)[37]. Structural similarity searches for the ZYG11B VHL box using Dali server against the Alphafold Database revealed that this extended loop is unique to the ZYG11 protein family, with the exception of ZYG11 from Caenorhabditis elegans (Supplementary Fig. 5b). The LRR domain constitutes of ten α-helix/β-strand repeats and adopts a curved solenoid conformation. In this configuration, ten α-helices align along the convex surface, while ten parallel β-strands define the concave groove (Fig. 2e). The ARM domain is composed entirely of α-helices and interconnecting loops that fall into nine ARM repeats. The α-helices maintain the overall architecture, while the loops introduce variations in inter-repeat angles and orientations (Fig. 2f). Particularly, a long flexible linker between ARM7 and ARM8 prevents the whole ARM domain from folding into a continuous superhelical cylindrical structure, allowing ARM8 and ARM9 to swing back and shape the substrate binding pocket. This configuration is observed in our full-length structure and is consistent with previously reported crystal structures of the truncated ARM domain lacking ARM1-3 (Fig. 2f)[29]. Meanwhile, structural alignment also reveals a subtle conformational difference in ARM4 relative to our full-length structure, which may reflect the effects of protein truncation or crystal packing (Supplementary Fig. 5c).

### Structural basis of ZYG11B recruiting EloB-EloC and substrate

In the complex, EloB-EloC is recruited to the N-terminal VHL box of ZYG11B, while the twisted C-terminal ARM domain folds back to engage EloB, together resembling a bent 'Mobius strip' like conformation (Fig. 3a). ZYG11B interacts with EloB-EloC through three distinct interfaces involving extensive hydrogen bonds and hydrophobic contacts (Fig. 3b). The first interface (interface 1) is located at the VHL box, where EloC tightly associates with the BC box of ZYG11B, forming a large contact area. This interaction mode resembles those observed in other CRL2 substrate receptors, such as the von Hippel-Lindau factor and FEM1B protein (Supplementary Fig. 5d)[38]. Specifically, L18 of the BC box inserts into a hydrophobic pocket on EloC formed by residues V73, Y76, F93, I95, L103, A107, and L110 (Fig. 3c), while L104 of EloC fits into a hydrophobic cavity on the opposite side, composed of ZYG11B residues L19, C22, L23, L26, F50, P51, V54, and L58 (Fig. 3d). Additionally, two smaller flanking hydrophobic surfaces contribute to this interface: one involves BC box residue I21 and EloC residues Y76, Y79, F93, and I95; the other includes BC box residues L26, F33, E46, and F50 together with EloC residues L101 and M105 (Fig. 3e). Hydrogen bonds between A13, S14, Y16, L18 of the BC box and N85, Y76 of EloC further stabilize the interface (Fig. 3f). Two additional interaction surfaces, referred to as interface 2 and interface 3, further stabilize the complex. Interface 2 comprises residues A397 and R436 of the ARM3 domain contacting EloC residue S87, while interface 3 involves ARM9 residues R728 and H729 electrostatically interacting

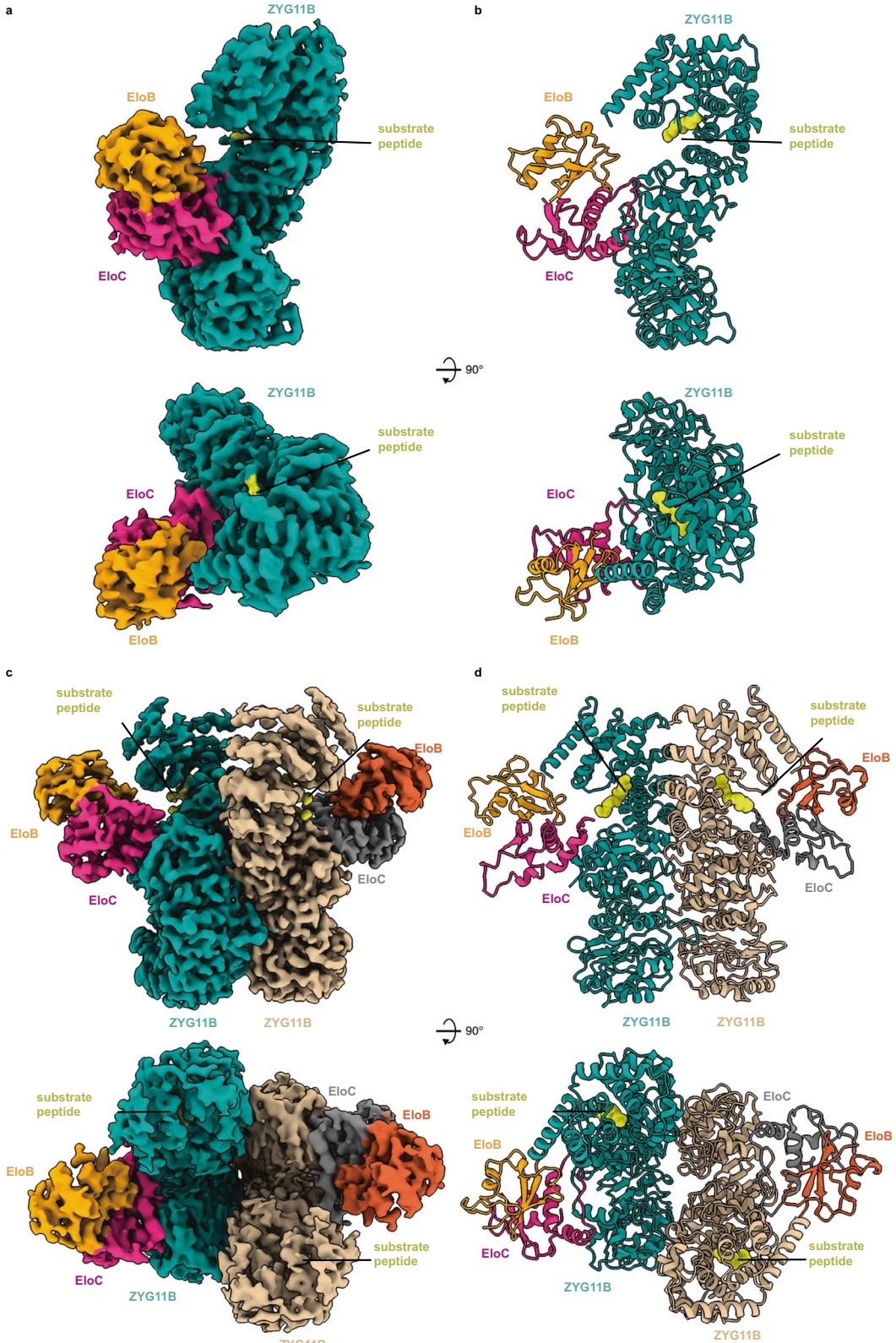

**Fig. 1 | Cryo-EM structures of monomeric and dimeric ZYG11B-EloB-EloC-substrate complexes.** Side (upper) and top (lower) views of cryo-EM maps (**a**, **c**) and models (**b**, **d**) for the ZYG11B-EloB-EloC-substrate monomer (**a**, **b**) and dimer (**c**, **d**).

with EloB residues E59—an interaction not observed in other CRL2 substrate receptors (Fig. 3g, h).

The degron peptide binds snugly within the groove of ARM domain, with its N-terminal glycine residue inserting into a substrate-binding pocket and its C-terminus exposed to solvent (Fig. 3a). This

binding mode is consistent with previous crystal structures of truncated ARM domains fused to degron peptides (Supplementary Fig. 5c)[27–29]. In our full-length structures, the first three residues of the degron are firmly anchored within the binding cavity through a main-chain mediated hydrogen-bonding network involving ZYG11B residues

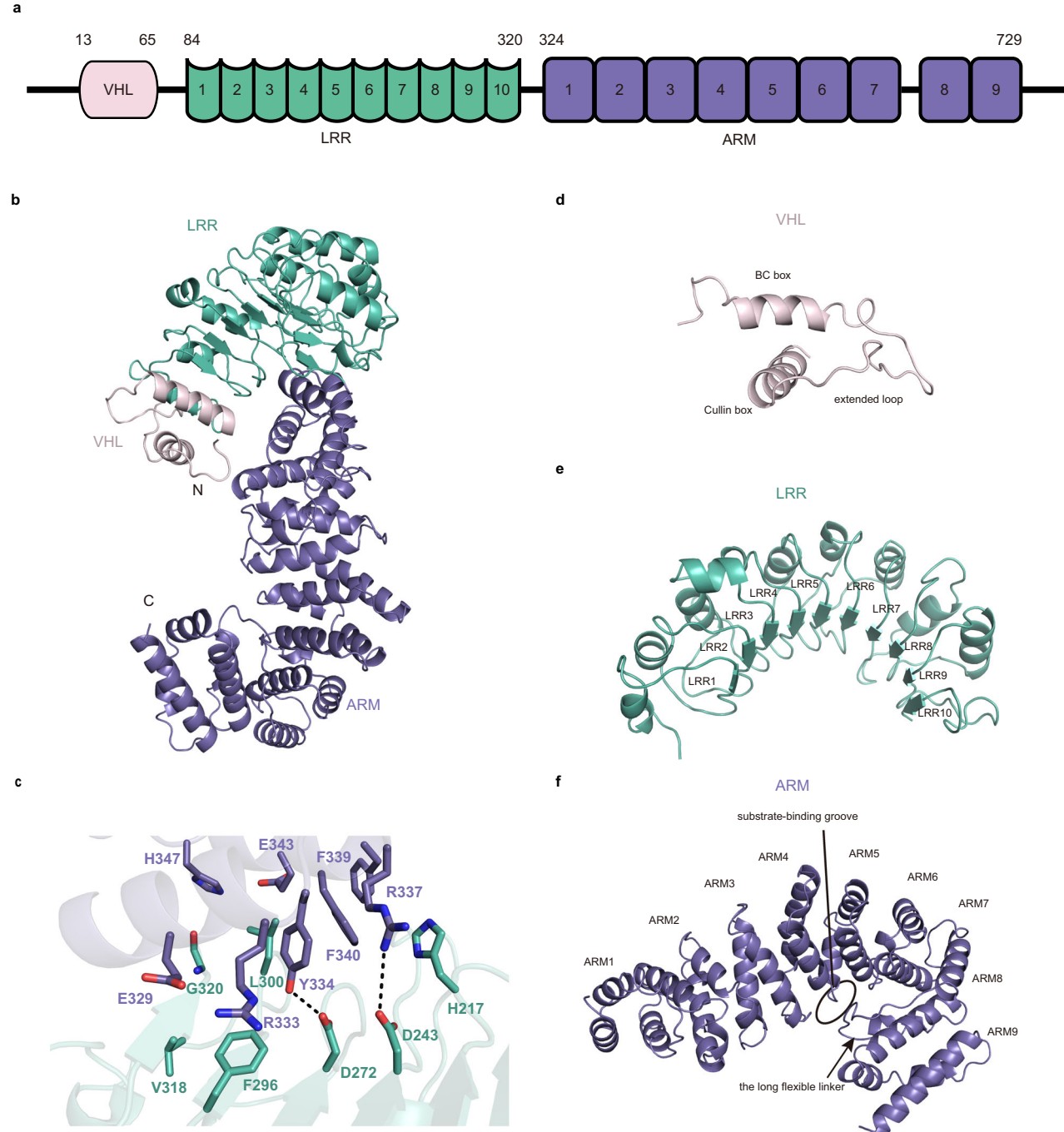

**Fig. 2 | Overall structure of ZYG11B. a** Domain organization of ZYG11. The VHL domain, LRR domain and ARM domain are shown in pink, cyan and purple, respectively. **b** Seahorse-shaped architecture of ZYG11B, colored as in (**a**). **c** Detailed hydrogen bonding and hydrophobic interactions between the LRR and ARM domains. Hydrogen bonds are shown as black lines. Structures of the VHL box (**d**), LRR domain (**e**) and ARM domain (**f**). The substrate-binding groove (ellipse) and long flexible linker (arrow) are indicated.

W522, N523, D526, N567, E570, and A647 (Fig. 3i). The side chains of Y2 and I3 from the degron form additional hydrogen-bonding and hydrophobic interactions with R649 and W522, respectively (Fig. 3j). Although the fourth residues of the degron remain buried within the pocket, no specific interactions with ZYG11B were observed in our structures (Fig. 3i).

To validate the functional importance of these interactions, we conducted mutagenesis studies and evaluated the substrate stability of the degron-fused proteins using the global protein stability system (GPS) in vivo (Fig. 3k)[20]. The degron sequence was derived from DBNDD2, a known endogenous substrate of ZYG11B. In this assay,

HEK293T cells were stably transfected with a vector encoding ubiquitin-(N-glycine-degron)-EGFP-P2A-DsRed. Upon translation, P2A-mediated cleavage produces separate ubiquitin-degron-EGFP and DsRed proteins. Following deubiquitination, the degron-EGFP is recognized by ZYG11B and directed to proteasome degradation, and the EGFP/DsRed ratio can reflect the stability of the substrate and the activity of ZYG11B. As expected, expression of wild-type ZYG11B remarkably reduced the EGFP/DsRed ratio through promoting degradation of the degron-fusion EGFP (Fig. 3l). In contrast, the W552A mutation within the substrate-binding pocket failed to promote degradation, whereas the empty vector condition exhibited slightly

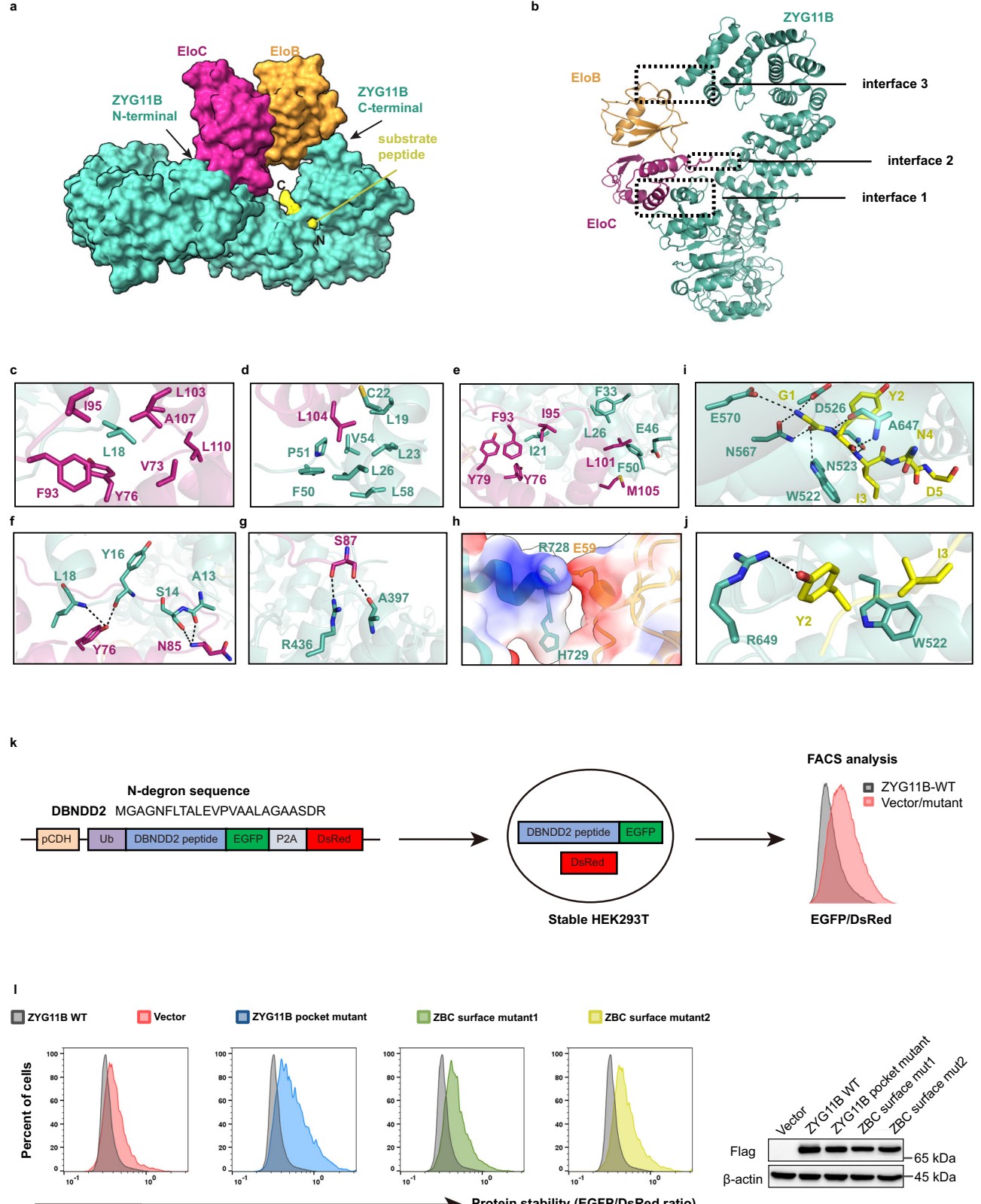

increased substrate stability than mutants, probably due to a dominant-negative effect. Moreover, mutants disrupting the EloC-interacting interface (mutant1: L18S; mutant2: P47A/V49A/F50A/P51A) retained high EGFP/DsRed ratios compared to wild-type ZYG11B, indicating impaired activity (Fig. 3l). Collectively, these results highlight the vital roles of specific ZYG11B residues in recruiting EloB-EloC and recognizing substrates.

## Structural details of ZYG11B-mediated dimerization

The dimeric assembly comprises two copies of the ZYG11B-EloB-EloC-substrate complex, forming a 2:2:2:2 stoichiometry with C2 symmetry. Within the dimer, the two ZYG11B molecules are arranged back-to-back, each recruiting an EloB-EloC complex and capturing a degron peptide, with the peptide C-termini oriented in opposite directions (Figs. 4a and 1c). All three domains of ZYG11B—the VHL box, LRR

**Fig. 3 | Interactions of ZYG11B with EloB-EloC and substrate peptide. a** Overall structure of the ZYG11B-EloB-EloC-substrate complex adopts a nicked Möbius strip–like conformation, shown in surface representation. **b** Three interaction surfaces between ZYG11B and EloB-EloC. **c–h** Detailed hydrogen bonding and hydrophobic interactions between ZYG11B (cyan) and EloB (orange)-EloC (magenta). Hydrogen bonds are shown as black lines. Detailed hydrogen bonding (**i**) and hydrophobic interactions (**j**) between ZYG11B (cyan) and substrate (yellow). Hydrogen bonds are shown as black lines. **k** Schematic of the cellular GPS assay.

pCDH lentiviral vector, Ub ubiquitin, EGFP enhanced green fluorescent protein, P2A porcine teschovirus-1 2A peptide, DsRed Discosoma red fluorescent protein. **l** Stability of DBNDD2 degron-fused EGFP in GPS reporter cells with exogenous expression of wild-type ZYG11B, pocket mutant (W522A), or EloB-EloC interacting surface mutants (mutant1: L18S, and mutant2: P47A/V49A/F50A/P51A) (left) and immunoblot of related proteins in GPS reporter cells (right). The experiment was repeated three times independently with similar results. Source data are provided as a Source Data file.

domain, and ARM domain—contribute to the formation of an extensive, tightly packed dimer interface, spanning ~1358 Å$^2$ (Fig. 4b). Specifically, Q38 from the protruding loop of the VHL box forms hydrogen bonds with E460, Q499, and Q503 of the opposing ARM domain (Fig. 4c). A second hydrogen-bonding network is organized by I93, S94, N116, and D118 of the LRR domain and N420 and H421 of the opposing ARM domain (Fig. 4d). Additionally, L150 from the LRR domain inserts into a hydrophobic pocket formed by L377, L381, and H421 of the opposing ARM domain, and further establishes a hydrogen bond between the main chain of L150 and the side chain of R342 (Fig. 4e). Finally, the two LRR domains also interact directly at two sites: one featuring a hydrogen bond between D152 and K248, together with hydrophobic contacts between Y175 and the backbone and side chains of Q249 and F250; and a second site involving a crosswise hydrogen-bond network formed by K220, L222, and K223 (Fig. 4f, g). Owing to the two-fold symmetry, these interactions are reciprocally mirrored on the opposite side of the dimer. Together, these interlocking interactions robustly stabilize the dimeric architecture.

To further investigate the full holoenzyme assembly, we reconstructed a low-resolution cryo-EM map of the hexamer complex containing ZYG11B-ELoB-EloC-Cul2-Rbx1 and the degron peptide (Supplementary Fig. 1b). By aligning our ZYG11B-ELoB-EloC-degron structure with the previously reported VHL-ELoB-EloC-Cul2-Rbx1 structure (PDB: 5N4W) via the VHL-EloB-EloC subunit, we built a composite model of the intact CRL2$^{ZYG11B}$ E3 ligase (Supplementary Fig. 6a). This model fits well into the cryo-EM map and positions the C-terminus of the degron peptide towards the catalytic subunit Rbx1, thereby providing a direct spatial basis for coupling substrate recognition with ubiquitin transfer (Supplementary Fig. 6a, b). We further built a dimeric model of the holoenzyme applying the same C2 symmetry observed in the ZYG11B-ELoB-EloC-degron dimer (Supplementary Fig. 6c). The resulting model exhibits no steric clashes and aligns with the 2D class averages (Supplementary Fig. 1b). Together, these analyses support the structural plausibility and biological relevance of a dimeric CRL2$^{ZYG11B}$ E3 ligase assembly.

To evaluate the functional importance of the dimeric assembly, we first established the in vitro ubiquitination assay to assess enzymatic activity. Given the extensive buried surface area of the dimer interface, a potential dimer-disrupting mutant was designed, incorporating combined substitutions containing L150S/D152S/R153S/K223A/F418A/P419A/N420A. The substrate contains an N-terminal ZYG11B-recognized degron peptide (GYFQRGK) which is followed by a long GS linker and a C-terminal GFP-FLAG tag[29]. As shown in the time-course biochemical assay (Supplementary Fig. 7a, b), wild-type CRL2$^{ZYG11B}$ promoted detectable mono- and di-ubiquitination of the substrate within 5 min, whereas the dimer-disrupting mutant showed weaker ubiquitination at all examined time points. Peptide-competition and enzyme concentration–gradient experiments further confirmed that substrate ubiquitination was dependent on CRL2$^{ZYG11B}$ E3 ligase activity (Supplementary Fig.7c, d). Moreover, in subsequent in vivo GPS assays, this mutant significantly impaired the degradation of degron-fused proteins compared with wild-type ZYG11B and other insufficient mutations (Fig. 4h and Supplementary Fig. 8a, b). These results indicate that dimerization of CRL2$^{ZYG11B}$ is critical for its E3 ligase activity and related biological functions.

## Discussion

Substrate receptors serve as key determinants of substrate specificity in CRL E3 ligases. ZYG11B and its paralog ZER1 act as central CRL2-associated receptors in the recently identified Gly/N-degron pathway, expanding the substrate repertoire and biological functions of the ubiquitin-proteasome system[20]. In this study, we resolved cryo-EM structures of the full-length ZYG11B-EloB-EloC complexed with a Gly/N-degron peptide in both monomeric and dimeric states, and proposed the structural model of CRL2$^{ZYG11B}$ holoenzyme, thereby elucidating the molecular architecture and assembly mechanism of this E3 ligase. A recent bioRxiv study reported a structure of CRL2$^{ZYG11B}$ in complex with a longer degron peptides derived from NLRP1 and ORF10, revealing the mode of substrate recognition and subunit interaction by monomeric CRL2$^{ZYG11B}$. However, a dimeric state was not observed in that study[39]. In contrast, we captured a dimeric assembly and demonstrated its functional relevance with both in vitro ubiquitination assays and in vivo GPS assays. Together, these complementary studies deepen our understanding of the molecular logic underlying substrate recognition and reveal the structural versatility of CRL2$^{ZYG11B}$ within the ubiquitin signaling landscape.

Our structure reveals that ZYG11B engages Gly/N-degron substrates through a compact substrate-binding groove within its C-terminal ARM domain. The N-terminal four residues of the degron peptide insert into a deep pocket and form an extensive interaction network with key residues of ZYG11B. This recognition mode is consistent with previous crystal structures of truncated ARM domains fused to degron peptides[27–29], but our full-length structure confirms that this interaction is preserved in a more native architecture, and further reveals how it is spatially integrated with adaptor recruitment and overall complex assembly. Mutagenesis results demonstrate that disruption of substrate-binding residues abolishes degron-dependent degradation in cells, underscoring the functional relevance of this recognition mechanism. Moreover, emerging evidence suggests that ZYG11B may employ alternative or extended binding surfaces—beyond the canonical substrate-binding groove—to engage non-canonical substrates or binding partners such as cyclin B1, or cGAS[29,36,39], although the structural basis for these interactions remains to be defined.

An unexpected defining feature of ZYG11B revealed by our study is its capacity to form symmetric homodimers. The dimeric assembly is stabilized by extensive inter-protomer interactions involving all three structural domains—VHL, LRR, and ARM—and positions two substrate-binding sites in opposite orientations. This configuration may facilitate multivalent substrate recognition, enhance the spatial coordination between substrate engagement and E2–Ub transfer, or increase the overall structural stability of the complex. Structural modeling indicates that dimerization is compatible with Cul2-Rbx1 binding and substrate ubiquitination, without introducing steric hindrance. Functionally, disrupting the dimer interface markedly impairs ZYG11B-mediated substrate degradation in cells, demonstrating that dimerization is not merely structural but mechanistically essential. Emerging studies have shown that oligomerization of CRL substrate receptors exerts diverse regulatory roles. For example, KLHDC2 and DCAF1 form autoinhibited higher-order assemblies, but become functionally active in the dimeric state; others, such as Fbw7, Cdc4, and FEM1B, display enhanced or selective activity upon dimerization; and SPOP assembles into higher-order oligomers that promote phase-separated substrate processing[30,32–35,40–43]. In this context,

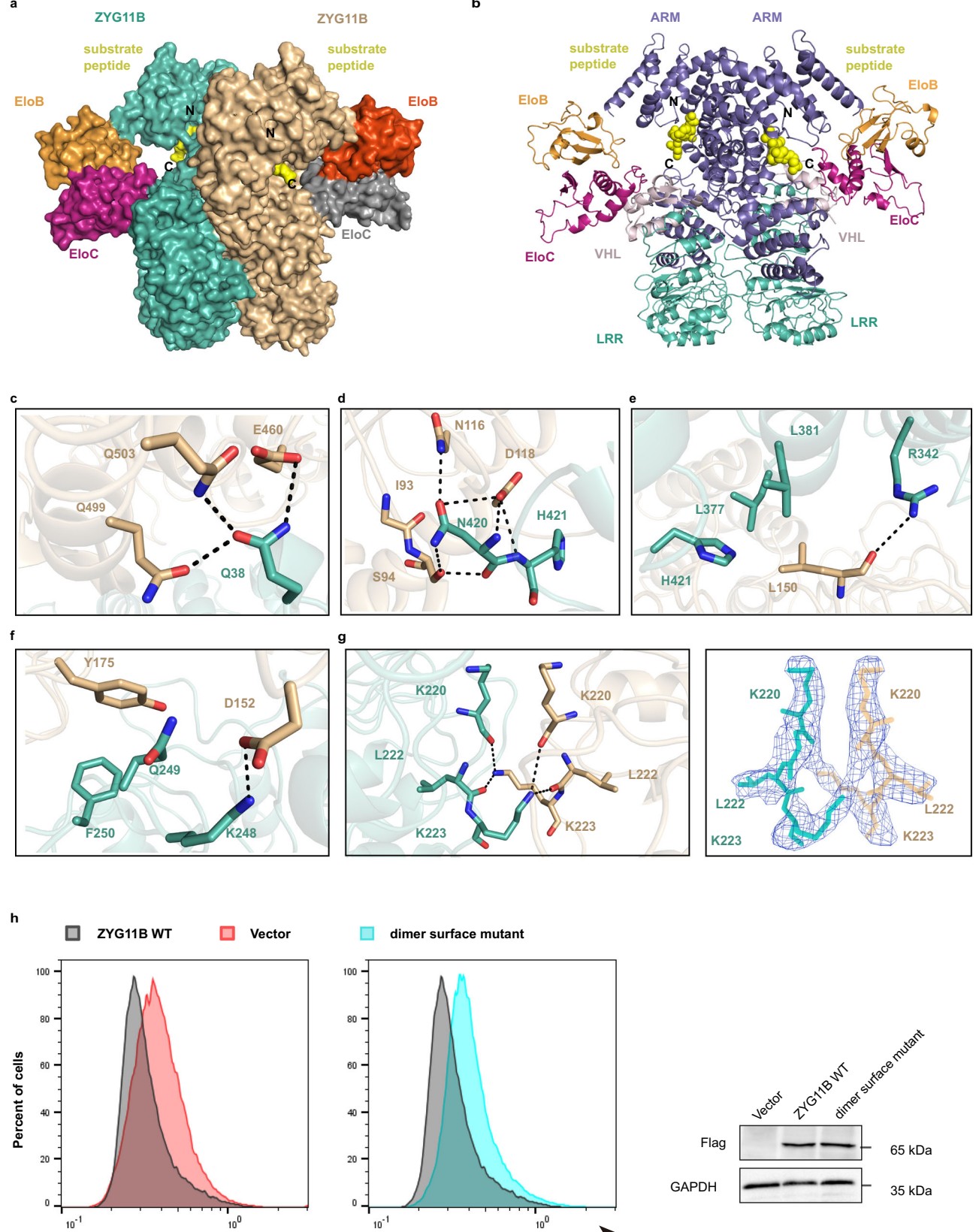

**Fig. 4 | Interactions between ZYG11B-mediated dimer surfaces.** Surface (**a**) and cartoon (**b**) views of the dimeric ZYG11B-EloB-EloC-substrate complex. Dimerization is mediated by ZYG11B, with the C terminus of substrate peptide oriented in opposite directions. **c–g** Detailed hydrogen bonds and hydrophobic interactions between ZYG11B dimer surfaces. Hydrogen bonds are shown as black lines.

**h** Stability of DBNDD2 degron-fused GFP in GPS reporter cells with exogenous expression of WT ZYG11B, and dimer surface mutant (L150S/D152S/R153S/K223A/F418A/P419A/N420A) (left), and immunoblot of related proteins in GPS reporter cells (right). The experiment was repeated three times independently with similar results. Source data are provided as a Source Data file.

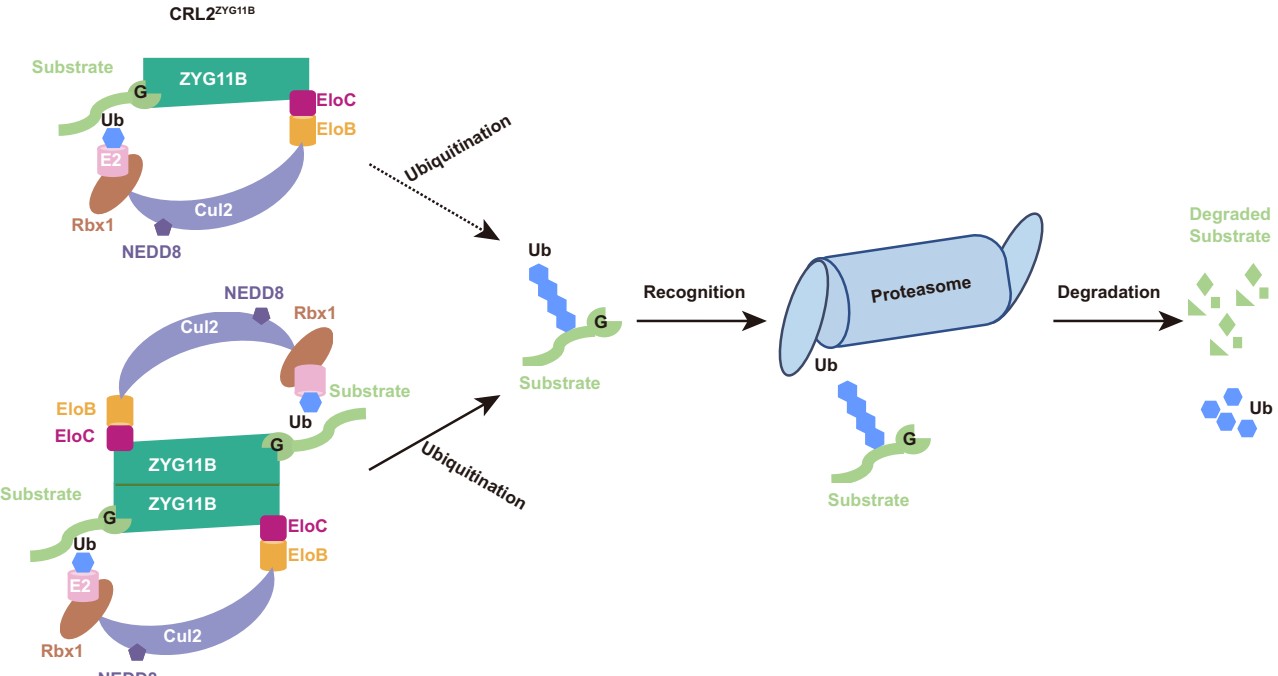

**Fig. 5 | Proposed working model for the CRL2<sup>ZYG11B</sup> E3 ligase.** Structural availability of the ZYG11B-mediated monomeric and dimeric states of CRL2<sup>ZYG11B</sup>, together with biochemical and functional assays, indicate that both states are required for substrate recognition, ubiquitination, and subsequent degradation by the UPS. The dashed line represents residual activity of monomer, while the solid line represents the demonstrable activity of dimer formation.

ZYG11B represents a dimer-forming receptor with demonstrable functional activity. However, the presence of monomeric assemblies in our structures raises the possibility that both states contribute to function. While our GPS assay results support a critical requirement for dimerization, we cannot definitively exclude residual activity of the monomeric form, given the intrinsic resolution limitations of cellular degradation assays. We therefore propose a working model in which CRL2<sup>ZYG11B</sup> cycles between monomeric and dimeric states, both of which contribute to substrate ubiquitination under cellular conditions (Fig. 5). This dynamic equilibrium may represent a regulatory mechanism for fine-tuning CRL2<sup>ZYG11B</sup> activity.

Furthermore, our high-resolution structures of full-length ZYG11B provide a valuable blueprint for developing ZYG11B-based PROTACs, where precise control of substrate engagement and complex stoichiometry is critical for therapeutic design. Firstly, ARM and LRR domains are known to mediate protein-protein interaction or provide favorable ligand-binding surfaces[44,45]. Accordingly, our high-accuracy atomic model offers a practical framework for structure-guided small-molecule design. Indeed, in addition to the substrate-binding pocket within the ARM domain, we identified an extended groove that reaches toward the dimer interface, which may serve as a potential ligand-binding site (Supplementary Fig. 9). Secondly, small molecules that stabilize the dimeric state of the CRL2<sup>ZYG11B</sup> complex or promote its monomer-to-dimer transition could represent an alternative PROTAC strategy, in contrary to KLHDC2, where degraders dissociate autoinhibited CRL2<sup>KLHDC2</sup> tetramers into activate dimers[46].

In summary, our findings establish a structural framework for understanding ZYG11B-mediated ubiquitination and provide mechanistic insights that may guide future therapeutic strategies leveraging CRL2<sup>ZYG11B</sup> as a programmable degradation platform.

## Methods
### Plasmid construction
The genes encoding the CRL2<sup>ZYG11B</sup> complex members, including full-length ZYG11B, EloB, EloC, Cul2, and Rbx1, were amplified from human complementary DNA (cDNA) using standard PCR methods. The ZYG11B coding sequence was cloned into a modified pFastBac Dual vector to express ZYG11B-TEV-GFP-6×His, or into pRSFDuet to express a His-SUMO-tagged ZYG11B. Human EloB and EloC were cloned into the modified pFastBac Dual vector or pACYC Duet vector at the first expression site (BamHI, HindIII) with a TEV-cleavable N-terminal His tag and at the second expression site (XhoI, KpnI) without a tag, respectively. The human Cul2 and Rbx1 genes were also cloned into the pFastBac Dual vector at cloning sites 1 (with an N-terminal His-tev tag) and 2 (without tag), respectively.

We made pCDH-blasticidine-ubiquitin-(N-degron)-EFGP-P2A-DsRED constructs for GPS-reporter cells using the CloneExpress one-step cloning kit (Vazyme). The N-terminal peptides of DBNDD2 were cloned into the GPS-reporter vector. Wild-type and mutant ZYG11B were cloned into the pCDH-cmv-puro vector with an N-terminal Flag tag.

For in vitro ubiquitination assays, expression constructs for all recombinant proteins were generated as follows. Ubiquitin (Ubi), UBE2R, and the substrate fusion GYFQRGK-(GS)<sub>20</sub>-GFP-FLAG were cloned into a modified pRSF vector encoding an N-terminal His–SUMO tag, generating pRSF-His-SUMO-Ubi, pRSF-His-SUMO-UBE2R, and pRSF-His-SUMO-GYFQRGK-(GS)<sub>20</sub>-GFP-FLAG. UBE2D3 was cloned into pET-28a with an N-terminal His tag followed by a TEV protease cleavage site (pET28a-His-TEV-UBE2D3). UBA1 was cloned into pFastDual with an N-terminal His–TEV tag (pFastDual-His-TEV-UBA1). All constructs were verified by Sanger sequencing.

### Protein expression and complex purification
The ZYG11B-EloB-EloC ternary complex was co-expressed in Sf9 cells (Thermo Fisher Scientific, cat. no. 11496015) by co-infection with P3 baculoviruses encoding ZYG11B and the EloB-EloC bicistron at a 2:1 ratio. Recombinant baculoviruses encoding the CUL2–Rbx1 complex and UBA1 were amplified in Sf9 insect cells. Sf9 cells at a density of 2.0–2.5 × 10<sup>6</sup> cells ml<sup>−1</sup> were infected with the indicated baculoviruses and cultured for 72 h at 27 °C. Cells were harvested by centrifugation

(2000 × $g$, 10 min). Expression vectors encoding ZYG11B-EloB-EloC, ubiquitin, UBE2R, UBE2D3, and the substrate GYFQRGK-GFP-FLAG were transformed into *E. coli* BL21(DE3) cells. Protein expression was induced with 0.3 mM IPTG and cultures were grown overnight at 16 °C.

For purification of ZYG11B-EloB-EloC proteins, cell pellets were resuspended and lysed by ultrasonication in lysis buffer containing 50 mM Tris-HCl (pH 7.5), 200 mM or 500 mM NaCl, 10 mM imidazole, 5% glycerol, 2 mM β-mercaptoethanol (β-ME), and 1 mM phenylmethylsulfonyl fluoride (PMSF). The recombinant proteins were initially purified by Ni$^{2+}$ affinity chromatography. The supernatant was loaded onto the Ni$^{2+}$ column; then the protein was washed with 50 mM imidazole and eluted with 500 mM imidazole in low salt (50 mM NaCl) or high salt (300 mM NaCl) buffer, respectively. The eluate was treated with TEV and ULP1 protease digestion overnight and loaded onto a Ni$^{2+}$ column to remove the tags. The protein was then subjected to size-exclusion chromatography using a Superdex 200 Increase 10/300 GL column (GE Healthcare), equilibrated with low-salt buffer (20 mM HEPES, pH 7.5, 50 mM KCl, 1 mM DTT) or high-salt buffer (20 mM HEPES, pH 7.5, 300 mM KCl, 1 mM DTT). Prior to cryo-EM grid preparation, the synthetic substrate peptide GYIND was incubated with the ZYG11B-EloB-EloC complex on ice for 30 min at a 10:1 (peptide: complex) molar ratio.

The Cul2-Rbx1 complex was purified using a similar protocol, with minor modifications. The final purification step involved gel filtration in 20 mM HEPES (pH 7.5), 150 mM KCl, and 1 mM DTT. The Cul2-Rbx1-ZYG11B-EloB-EloC pentamer complex was assembled by mixing the purified components and subjected to size-exclusion chromatography in 20 mM HEPES (pH 7.5), 150 mM KCl, and 1 mM DTT. The ZYG11B-EloB-EloC-Cul2-Rbx1 complex was supplemented with the synthetic substrate peptide GYIND at a 10:1 (peptide: complex) molar ratio and incubated on ice for 30 min before cryo-EM grid preparation. The neddylated Cul2Δ-Rbx1 (Cul2Δ: Δ117-134) protein used in in vitro ubiquitination assays was a kind gift from Dr. Chao Xu at University of Science and Technology of China[33].

For proteins used in in vitro ubiquitination assays, including ubiquitin, UBA1, UBE2R, UBE2D3, and the substrate GYFQRGK-GFP-FLAG, recombinant proteins were initially purified by Ni$^{2+}$-affinity chromatography. Affinity tags were removed by overnight digestion with TEV protease or ULP1, followed by a second round of Ni$^{2+}$-affinity chromatography to remove the cleaved tags and proteases. The final proteins were stored in buffer containing 50 mM Tris-HCl (pH 7.5), 500 mM NaCl, and 5% (v/v) glycerol.

## Analytical ultracentrifugation (AUC) assays
Sedimentation velocity analytical ultracentrifugation (SV-AUC) experiments were performed using a ProteomeLab XL-I analytical ultracentrifuge (Beckman Coulter, Brea, CA), equipped with an AN-60Ti rotor (4-holes). Prior to the experiments, an additional purification step of the ZYG11B-EloB-EloC complex was performed using size-exclusion chromatography in low-salt buffer (20 mM HEPES, pH 7.5, and 50 mM KCl) or high-salt buffer (20 mM HEPES, pH 7.5, and 300 mM KCl). Samples were loaded with 380 μL of ZYG11B-EloB-EloC at a concentration of about 1 mg/ml and 400 μL of the same purification buffer. SV-AUC experiments were carried out at 20 °C and 70,000 × $g$ using a continuous scan mode with a radial spacing of 0.003 cm. Scans were collected in 3 min intervals at 280 nm. The fitting of absorbance versus cell radius data was performed using SEDFIT software and continuous sedimentation coefficient distribution c(s) model, covering a range of 0–20 S.

## Cryo-EM specimen preparation and imaging
Three microliters (3 μL) of purified ZYG11B-EloB-EloC with the substrate peptide (1 mg/ml) were applied to glow-discharged Quantifoil R1.2/1.3 Cu grids. The grids were incubated for 2 s, blotted for 6 s at 4 °C and 100% humidity, and then plunge-frozen in liquid ethane using a Vitrobot Mark IV (Thermo Fisher Scientific). Data collection of all the cryo-EM datasets was performed on the 200 kV Talos Arctica electron microscope (FEI) equipped with K2 summit camera (Gatan). All cryo-EM micrographs were collected automatically under super-resolution mode, yielding an image stack with a pixel size of 0.8 Å (×165,000). Images were recorded at a defocus range of −1.2 μm to −1.8 μm, with a total electron dose of 60 e$^-$ Å$^{-2}$ over 32 movie frames.

## Image processing
A total of 1987 micrographs were collected for ZYG11B-EloB-EloC-substrate dataset. Movies frames were aligned using Patch Motion Correction, and CTF parameters were estimated using Patch CTF Estimation in cryoSPARC. In total, 690,827 particles were picked and extracted. Multiple rounds of 2D classification yielded 347,626 particles with clear features, which were subjected to Ab initio reconstruction. After Ab initio reconstruction, two well-defined classes corresponding to monomeric and dimeric ZYG11B-EloB-EloC complexes were selected for further processing. For the monomer dataset, non-uniform refinement followed by local refinement generated a 3.37 Å map from 99,806 particles. For the dimer dataset, two rounds of non-uniform refinement with C2 symmetry produced a 3.27 Å map from 100,431 particles.

## Model building and refinement
The sharpened maps were generated in cryoSPARC. An initial model of the ZYG11B-EloB-EloC-substrate complex was built based on the AlphaFold-predicted structure of ZYG11B (AF-Q9C0D3-F1), together with the crystal structure of EloB-EloC (PDB ID: 5N4W) and the ORF10 peptide (PDB ID: 7XV7). The model was fit into the cryo-EM maps using UCSF Chimera and further manually adjusted in Coot. The resulting model was refined in PHENIX by real-space refinement with secondary-structure restraints. The final models were subjected to multiple rounds of real-space refinement in PHENIX and manual adjustment in Coot until convergence, with no further improvement observed. Model stereochemistry was validated using the comprehensive cryo-EM validation tools in PHENIX. Refinement statistics are summarized in Supplementary Table 1. Structural figures were prepared using PyMOL or ChimeraX.

## Cell culture and Lentivirus packaging
Human HEK293T cells (ATCC, CRL-3216) were cultured in Dulbecco's Modified Eagle Medium (DMEM; Biological Industries, Cat. No. 01-052-1ACS) supplemented with 10% fetal bovine serum (FBS; Biological Industries, Cat. No. 04-001-1ACS) and 1% penicillin-streptomycin (HyClone) at 37 °C in a 5% CO$_2$ incubator. For lentiviral packaging, transfer vector constructs (pCDH-ub-DBNDD2 peptide-EGFP-P2A-DsRed and pCDH-ZYG11B-WT or mutant), along with packaging plasmids psPAX2 and pCMV-VSV-G, were co-transfected into HEK293T cells. Forty-eight hours post-transfection, the lentivirus was harvested from the supernatant and filtered through a 0.45 μm filter (Millipore).

## Global protein stability assay
The GPS assay using the Ub-DBNDD2 peptide GPS reporter cell line was previously described (Xiaojie Yan et al. 2022). Briefly, the 19-mer DBNDD2 peptide was encoded as an oligonucleotide and cloned into the GPS vector. To generate GPS reporter cells, the GPS vector was packaged into lentivirus and transduced into HEK293T cells, followed by selection with blasticidin (30 μg/mL). To assess ZYG11B activity, lentiviral vectors encoding wild-type or mutant ZYG11B with an N-terminal Flag tag were packaged and transduced into GPS reporter cells, followed by selection with puromycin (10 μg/mL). Fluorescence signals of EGFP and DsRed were measured by flow cytometry using an LSR Fortessa instrument (Becton Dickinson). For the flow cytometry gating strategy, cells were first gated on FSC-A and SSC-A to exclude

debris and doublets. DsRed-positive cells were then selected as events with DsRed fluorescence intensity >10³, representing cells carrying the lentivirally integrated GPS reporters. GFP/DsRed fluorescence ratios were analyzed with FlowJo software and used to quantify the stability of the GFP-fused substrate. In the resulting distributions, the y-axis represents the normalized frequency of cells. Cells were also analyzed by Western blotting. The following primary antibodies were used: anti-FLAG antibody (mouse monoclonal, ABclonal, Cat# AE005; 1:2000), anti-GAPDH antibody (mouse monoclonal, ABclonal, Cat# AC033; 1:5000), and anti-β-actin antibody (mouse monoclonal, ABclonal, Cat# AC004; 1:5000). HRP-conjugated goat anti-mouse IgG (H + L) (ABclonal, Cat# AS003; 1:10000) was used as the secondary antibody.

### In vitro ubiquitination assays

For in vitro ubiquitination assays, UBA1 was used as E1 enzyme, and two E2 enzymes (UBE2R1 and UBE2D3) combination were used to enhance efficiency according to previous reports[32]. The assays were performed at 37 °C in reaction buffer (50 mM HEPES, pH 7.5, 100 mM NaCl, 10 mM MgCl₂) containing 1 μM UBA1, 1 μM UBE2R1, 1 μM UBE2D3, 1 μM neddylated Cul2Δ-Rbx1, indicated amounts of wild-type ZYG11B-EloB-EloC or mutants, 1 μM substrate GYFQRGK-GFP-FLAG, and 100 μM ubiquitin. 5 mM ATP was added to the mixture to initiate the reaction. For competition experiments, degron substrate peptide (GYFQRGK) was added to the mixture at indicated concentrations. Reaction systems were mixed with SDS-PAGE loading buffer after indicated time and boiled for 10 min to terminate the reaction. Ubiquitinated products were detected by immunoblotting using an anti-GFP antibody (mouse monoclonal, ABclonal, Cat# AE012; 1:2000 dilution), followed by an HRP-conjugated goat anti-mouse IgG (H + L) secondary antibody (ABclonal, Cat# AS003; 1:10000 dilution).

### Reporting summary

Further information on research design is available in the Nature Portfolio Reporting Summary linked to this article.

## Data availability

The atomic coordinates and cryo-EM density maps have been deposited in the Protein Data Bank (PDB) and Electron Microscopy Data Bank (EMDB) under accession codes: PDB: 9LK2 and EMDB: EMD-63161 for monomeric ZYG11B-EloB-EloC with substrate peptide GYIND, PDB: 9LK6 and EMDB: EMD-63169 for dimeric ZYG11B-EloB-EloC with substrate peptide GYIND, PDB: 5N4W for Cul2-Rbx1-EloBC-VHL complex (pre-release), PDB: 7XV7 for ZYG11B bound to ORF10 peptide. Source data are provided with this paper.

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

## Acknowledgements

Cryo-EM data collection was carried out at the Center for Biological Imaging, Core Facilities for Protein Science at the Institute of Biophysics, Chinese Academy of Sciences. Computation work was performed using the high-performance computing resources at the Center for Biological Imaging, Institute of Biophysics, Chinese Academy of Science. We thank Cryo-Electron Microscopy Platform of Medical Science and Technology Innovation Center of Shandong First Medical University for the support of Cryo-EM data collection. We thank Xiaojun. Huang, Boling. Zhu, Xujing. Li, Fei Sun, and other staff members at the Center for Biological Imaging for their support in data collection; Xiaoyan. Wang for helping with FACS; Xiaoxia Yu and Qian Wang for their support with the analytical ultracentrifugation. We thank Dr. Chao Xu and Jiayu Wang at the University of Science and Technology of China for kindly providing the neddylated Cul2Δ-Rbx1 protein. We thank the help of Dr. Likun Wang, Si Chen and Tao Li for the generation of cell lines used in the GPS assay. This work was supported by grants from National Science and Technology Major Project of China (2025ZD01904303 to P.G.), Major Project of Guangzhou National Laboratory (GZNL2024A01011 to P.G.), National Key R&D Program of China (2024YFA1307400 to P.G.), National Natural Science Foundation of China (32325028 & 32130057 to P.G., 32071200 to Y.W., 32271266 to H.F.), Beijing Natural Science Foundation (Z220018 to P.G.), CAS Project for Young Scientists in Basic Research (YSBR-074 to P.G.), and Natural Science Foundation of Shandong Province (ZR2024QC360 to N.L. and ZR2024QC358 to M.S.).

## Author contributions

N.L. purified proteins, prepared cryo-EM samples, collected and processed cryo-EM data, reconstructed density maps, cellular assays and biochemical assay. Y.S.G. processed cryo-EM data and reconstruction. Y.G. assisted with cryo-EM data process and reconstruction. Y.G., H.F., M.S., Y.W. and P.G. assisted with built and refined models. S.Q.L. assisted with cell culture and protein expression. Y.W. and P.G. initiated the project and directed the research. Y.W., N.L. and P.G. wrote the manuscript with the help of all of the authors.

## Competing interests

The authors declare no competing interests.
