## [Transparent Peer Review file · Nature Communications]

Structures of ZYG11B-EloB-EloC-substrate complex reveal mechanisms of CRL2^{ZYG11B} assembly and function

Corresponding Author: Professor Pu Gao

Version 0:

Reviewer comments:

Reviewer #1

(Remarks to the Author)

The manuscript by Lin et al. reports very nice cryoEM-derived structures of full-length ZYG11B in complex with other components of this interesting E3 ligase plus a substrate-derived peptide. Prior structures used fragments of ZYG11B. While the current manuscript presents compelling structural biology, functional assays to support implications of the structural findings are limited or have caveats. Below are my specific comments, in approximate order of importance.

1) Figure 4g: I commend the authors for providing the immunoblots to show relative levels of the ZYG11B variants in cells. My concern is that the “dimer surface mutant” was at significantly lower levels than the WT protein. What would the data/plot look like if WT levels were matched with the mutant?

Even if ZYG11B variants were matched in levels by blot, their cellular localizations or folding quality could be different, especially given the high number of amino acid substitutions for the dimer surface mutant and overexpression. CRL substrate recognition subunits have long been known to not fold properly in the absence of their binding partners and specific chaperones (Feldman et al., PMID:10635329; Taipale et al., PMID:25036637). The authors would need to demonstrate both equal levels plus equal functionality in some way, such as equal co-IP of EloB/C.

An alternative is to use biochemistry to investigate E3 activity (see comment #2).

2) Supplemental Figure 5 shows nice data for the presumed full E3 complex: CUL2/Rbx1/EloB/EloC/ZYG11B. I write “presumed” because I have not seen any report of reconstituted activity for this E3, with a peptide substrate for example. If the authors could develop such an assay, it would greatly enhance the current manuscript and allow for more controlled functional tests of their structure-based hypotheses. I suggest using the DBNDD2-derived peptide with added lysine residues and a tracer near the C-terminus. See Hickey et al. (PMID:38177675) for an analogous assay for CRL2-KLHDC2.

Can the dimer surface mutant be purified? If so, is it less active than the dimer? Only about 2-fold less active? Does it bind EloB/C and substrate peptide? Other mutants could also be studied, but I think the implications of dimerization are most pressing—and novel—for the current manuscript.

There are also some interesting opportunities with a biochemical activity assay and purified mixed ZYG11B dimers (would require two different tags). For example, would a dimer with two (non-mutated) active sites be only twice as active as a dimer with only one active site?

Combined, these additional studies may suggest that the main reason ZYG11B dimerizes is for enhanced protein stability. More interesting findings are also possible, though from the current manuscript I only see evidence for dimerization creating two identical substrate-binding pockets that could likely function in monomers. The authors even present a model in which monomers and dimers contribute to E3 activity. Biochemical activity assays could provide significant resolution.

3) The manuscript discussion mentions “a valuable framework for the development of ZYG11B-based PROTACs”. I did not find obvious ways that the manuscript would aid in PROTAC design. Again, I see two identical substrate-binding pockets consistent with previous reports. Does the dimer interface create another potentially ligandable site? For KLHDC2,

tetramerization might explain relatively weak PROTACs (or ligands for PROTACs) if the PROTACs/ligands are not capable of disrupting the tetramer, as the peptides do. From the current manuscript, it does not appear that ZYG11B dimerization would present a problem or advantage for PROTACs. Please elaborate if there are specific ideas.

4) Supplemental Figure 1 (and methods): It's not clear to me which conditions had the substrate peptide. Parts A through C of the figure? Did the authors perform experiments to determine whether the substrate peptide influences the monomer-dimer state?

Minor comments:

5) Methods, line 275: please provide more detail on the pCDH-puro vector. CMV promoter? There are many versions of this vector. The promoter in the plasmid matters since the level of protein overexpression in cells can have important implications.

6) Methods, line 266: "C-terminal HIS-GFP-TEV tag" is confusing to me. It is best to list proteins in N to C order. So, was it ZYG11B-TEV-GFP-6HIS, leaving no tag behind after cleavage?

7) Figure 5: The proteasome cartoon would be better if made simpler (just one shape) instead of appearing as 10 stacked hexamers, which is not accurate. Or, perhaps Nature Publishing can provide a better proteasome cartoon that more accurately depicts the lid, base, and CP parts of the proteasome. Also, the color used for Ub makes it difficult to see above the background.

Reviewer #2

(Remarks to the Author)

This manuscript presents structural and functional analyses of the human CRL2^{ZYG11B} E3 ubiquitin ligase, focusing on how ZYG11B engages substrates bearing Gly/N-degrons. Using cryo-electron microscopy, the authors report structures of full-length ZYG11B bound to the EloB–EloC adaptor complex and a Gly/N-degron peptide in both monomeric and dimeric assemblies. Complementary cell-based assays support the notion that disrupting substrate, adaptor, or dimerization interfaces impairs substrate degradation. While the study offers some new observations, there are several major concerns that should be addressed to clarify the novelty and rigor of the findings.

First, a closely related study was posted on bioRxiv in 2024 and is only briefly mentioned in the Discussion. The authors do not adequately acknowledge that the previously reported structures overlap significantly with those presented here. This raises concerns about the originality of the work and diminishes its novelty. The manuscript should provide a more transparent comparison to the bioRxiv study and clearly articulate what is uniquely contributed here.

Second, the reported ZYG11B dimer represents a potential novel finding not described in the bioRxiv paper. However, the biological relevance of this dimer is not convincingly established. Specifically, the manuscript lacks structural data of the full CRL2^{ZYG11B} complex including Cul2, which would be necessary to demonstrate that the dimeric form is retained upon assembly with the full E3 ligase machinery. The only supporting evidence—weak density corresponding to Cul2 in 2D class averages—is not sufficient. Stronger experimental validation, such as reconstruction of the dimeric complex with Cul2-Rbx1, is needed to substantiate this claim.

Third, the structural data are not presented with sufficient clarity or transparency to allow readers to evaluate the quality of the reconstructions. The reported resolutions (~4 Å) are modest and may not support unambiguous side chain assignments, yet the manuscript contains detailed interaction descriptions (e.g., hydrogen bonding networks) that may go beyond what is justified by the data. With the exception of Fig. 1, cryo-EM density maps are not shown in the main figures. Furthermore, figures such as 2C, 3a, 3c–e, 3j, and 4e–f appear to use surface renderings derived from atomic models, not the experimental maps themselves, which could mislead readers about the level of structural detail actually supported by the data. The authors state that AlphaFold models were used for initial model building, but it remains unclear to what extent the final structural conclusions rely on experimental evidence versus predicted models.

Fourth, the GPS degradation assays are insufficiently described and potentially misleading. The presentation of the data lacks normalization details; the Y-axis scale and its relation to population-level distributions are unclear. Mutant constructs show effects beyond the vector control, which suggests potential dominant-negative behavior that is not explored or acknowledged. The influence of variable expression levels, as seen in Western blots, is not discussed. Moreover, the choice of mutations—particularly the combination of L150S/D152S/R153S/K223A/F418A/P419A/N420A—seems arbitrary and is the only construct used to support the functional relevance of dimerization. Additional mutants or rescue experiments would be helpful to strengthen this conclusion.

Minor points:

Line 101: The term "electron density" is not appropriate in the context of cryo-EM and should be replaced with "electrostatic potential map" or "Coulomb potential."

The title is somewhat misleading; the study focuses on the ZYG11B–EloB–EloC complex and substrate recognition. The involvement of the full CRL2 ligase (including Cul2 and Rbx1) is limited and not structurally characterized in depth.

In summary, while the manuscript contains potentially interesting observations, the novelty relative to prior work, the

interpretation of the structural data, and the functional validation all require clarification or strengthening to meet the standards of a high-impact journal.

Version 1:

Reviewer comments:

Reviewer #1

(Remarks to the Author)

The authors nicely addressed my comments from the original review. I will only comment further on the cell-free ubiquitination experiments (new data; Supplemental Figure 7). The legend describes the ubiquitination as “robust”. It is not robust so I suggest alternative language. Sometimes these cell-free reactions are weak but if the controls are consistent with E3-dependent ubiquitination (see below), I'd deem these studies acceptable. A couple of notes/suggestions that could enhance cell-free activity. 20 mM ATP is a high concentration—and ATP can bring down the pH of the final reaction substantially if the ATP stock is not pH adjusted or buffered enough. I'd suggest testing 2 mM ATP final and be mindful of the pH. One could also adjust the concentrations of the enzymes and substrate. Also, how were the specific ubiquitin conjugating enzymes chosen? Add a short commentary and cite Li et al. 2024 (PMID: 38382526) or another reference.

In the Methods section, the authors write that “reaction mixtures without ATP were used as control”, but I do not see such data. This control, and ideally a control with excess substrate peptide to occupy that pocket on the E3, would strengthen the findings. The “no ATP” condition should also show a drastic difference on an anti-ubiquitin blot but the substrate peptide should not affect ubiquitination of non-substrate components (ubiquitination of E3 components is common in these types of reactions) but should reduce your model substrate ubiquitination. An alternative to the excess substrate peptide control would be to leave out the E3. Sometimes 1 or 2 ubiquitins can be added to substrates without an E3 in these types of reactions involving extremely high concentrations of everything. Better overall E3-dependent substrate ubiquitination would allow comparison of multiple concentrations of WT and mutant E3 to estimate how much dimerization impacts E3 activity.

Minor comment: While “ubiquitination” and “ubiquitylation” are interchangeable terms, it is good practice to use one consistently in a single article.

Reviewer #2

(Remarks to the Author)

I am satisfied with the authors' revisions, which substantially improve the rigor and clarity of the manuscript. The additional structural and biochemical are largely convincing.

One more point: While the rebuttal explains why a multi-site mutant was necessary to abolish the dimer interface, this should be briefly stated in the manuscript (better with supplemental data) to clarify why single or limited mutations were insufficient.

Version 2:

Reviewer comments:

Reviewer #1

(Remarks to the Author)

I'm satisfied with the revised manuscript. Nice work!

Overall Response:

We sincerely thank the reviewers for their careful evaluation and constructive comments. In response, we have substantially revised the manuscript and performed additional structural, biochemical, and cellular experiments to directly address the concerns raised. The major experimental additions and analyses are summarized below:

(1) We reprocessed the cryo-EM datasets, improving the resolution of the monomeric state from 3.95 Å to 3.37 Å and the dimeric state from 3.88 Å to 3.27 Å. These improvements enable more reliable interpretation of intermolecular interactions.

(2) We performed analytical ultracentrifugation (AUC), providing independent solution-based biochemical evidence for the coexistence of monomeric and dimeric assemblies of the CRL2^{ZYG11B} holoenzyme (ZYG11B-EloB-EloC-Cul2-Rbx1), consistent with cryo-EM 2D classifications.

(3) Cellular localization assays demonstrate that both wild-type ZYG11B and the dimer-interface mutant localize predominantly to the cytoplasm. In parallel, purified CRL2^{ZYG11B} complexes containing either wild-type or mutant ZYG11B co-elute as a single peak by size-exclusion chromatography, indicating intact assembly with EloB-EloC-Cul2-Rbx1.

(4) When expressed at comparable levels with wild-type ZYG11B, disruption of the dimer interface significantly impairs degradation of degron-fused substrates in cells, as assessed by GPS assays.

(5) We established a reconstituted *in vitro* ubiquitination assay using purified proteins. In this system, the dimer-interface mutant exhibits markedly reduced CRL2^{ZYG11B}-mediated substrate ubiquitination, providing direct biochemical evidence for the functional relevance of ZYG11B dimerization.

Together, these additions substantially strengthen the structural rigor and functional interpretation of the study and directly address the reviewers' concerns. Detailed point-by-point responses are provided below (reviewers' comments are shown in red, and our responses in black). Major changes in the revised manuscript are highlighted as underlined text.

Point-by-Point Response:

Reviewer #1 (Remarks to the Author):

The manuscript by Lin et al. reports very nice cryoEM-derived structures of full-length ZYG11B in complex with other components of this interesting E3 ligase plus a substrate-derived peptide. Prior structures used fragments of ZYG11B. While the current manuscript presents compelling structural biology, functional assays to support implications of the structural findings are limited or have caveats.

We thank the reviewer for the positive assessment of our cryo-EM structures and for highlighting the need for stronger functional validation. In response, we have performed additional in cells and *in vitro* experiments, including matched-expression GPS assays, biochemical reconstitution of ubiquitination activity, AUC analysis, and complex assembly controls. These data directly strengthen the functional implications of our structural findings and have been incorporated into the revised manuscript.

Below are my specific comments, in approximate order of importance.

1) Figure 4g: I commend the authors for providing the immunoblots to show relative levels of the ZYG11B variants in cells. My concern is that the “dimer surface mutant” was at significantly lower levels than the WT protein. What would the data/plot look like if WT levels were matched with the mutant?

We thank the reviewer for pointing out this important issue. We agree that unequal expression levels could confound interpretation. To address this concern, we re-established stable cell lines in which wild-type ZYG11B and the dimer-interface mutant are expressed at comparable levels. Under these conditions, the dimer-interface mutant still shows a pronounced defect in promoting degradation of degron-fused substrates in the GPS assay (below figure and Fig 4h).

Even if ZYG11B variants were matched in levels by blot, their cellular localizations or folding quality could be different, especially given the high number of amino acid substitutions for the dimer surface mutant and overexpression. CRL substrate recognition subunits have long been known to not fold properly in the absence of their binding partners and specific chaperones (Feldman et al., PMID:10635329; Taipale et al., PMID:25036637). The authors would need to demonstrate both equal levels plus equal functionality in some way, such as equal co-IP of EloB/C.

Thanks. We agree with this concern and therefore performed additional controls.

First, by transiently expressing C-terminal GFP fused ZYG11B in HEK293T cells after 24 h post-transfection, wild-type and dimer-interface mutant ZYG11B exhibit indistinguishable cytoplasmic localization (below figure).

Second, purified CRL2^{ZYG11B} complexes containing either WT or mutant ZYG11B co-elute as a single peak by size-exclusion chromatography, indicating preserved interaction with EloB-EloC-Cul2-Rbx1 (below figure)

Together, these results indicate that the functional impairment observed for the dimer-interface mutant reflects disruption of dimerization rather than altered expression, folding, or complex assembly.

An alternative is to use biochemistry to investigate E3 activity (see comment #2).

2) Supplemental Figure 5 shows nice data for the presumed full E3 complex: CUL2/Rbx1/EloB/EloC/ZYG11B. I write “presumed” because I have not seen any report of reconstituted activity for this E3, with a peptide substrate for example. If the authors could develop such an assay, it would greatly enhance the current manuscript and allow for more controlled functional tests of their structure-based hypotheses. I suggest using the DBNDD2-derived peptide with added lysine residues and a tracer near the C-terminus. See Hickey et al. (PMID:38177675) for an analogous assay for CRL2-KLHDC2. Can the dimer surface mutant be purified? If so, is it less active than the dimer? Only about 2-fold less active? Does it bind EloB/C and substrate peptide? Other mutants could also be studied, but I think the implications of dimerization are most pressing—and novel—for the current manuscript.

We thank the reviewer for this constructive suggestion. Following this advice, we established a reconstituted *in vitro* ubiquitination assay using purified components, including E1, E2, ubiquitin, ATP/Mg²⁺, and neddylated Cul2-Rbx1 assembled with ZYG11B-EloB-EloC. The substrate contains an N-terminal ZYG11B-recognized degron (GYFQRGK) followed by a flexible linker and a C-terminal GFP-FLAG tag.

In this fully reconstituted system, wild-type CRL2^{ZYG11B} catalyzes robust mono- and di-ubiquitination of the substrate, whereas the dimer-interface mutant exhibits markedly reduced ubiquitination activity (below figure and Supplementary Fig.7a, b). Importantly, the mutant complex assembles normally with other CRL2 components, as assessed by SEC, indicating that the reduced activity reflects a functional defect rather than impaired complex assembly.

There are also some interesting opportunities with a biochemical activity assay and purified mixed ZYG11B dimers (would require two different tags). For example, would a dimer with two (non-mutated) active sites be only twice as active as a dimer with only one active site? Combined, these additional studies may suggest that the main reason ZYG11B dimerizes is for enhanced protein stability. More interesting findings are also possible, though from the current manuscript I only see evidence for dimerization creating two identical substrate-binding pockets that could likely function in monomers. The

authors even present a model in which monomers and dimers contribute to E3 activity. Biochemical activity assays could provide significant resolution.

We agree that biochemical analysis of heterodimeric complexes containing one WT subunit and one catalytically inactive subunit would be informative. We therefore attempted to generate such mixed complexes by co-expressing differently tagged ZYG11B variants together with EloB-EloC. Despite extensive optimization across multiple parameters (cell strains, medium, and induction conditions), we were unable to reproducibly isolate stable heterodimeric complexes. As a result, we could not obtain samples with sufficient stability and homogeneity to support quantitative biochemical analysis, and therefore did not include these experiments.

Importantly, in the revised manuscript we have added new biochemical and cellular assays that directly demonstrate that ZYG11B dimerization enhances E3 ligase activity. We believe that these new data sufficiently address the central functional question regarding the role of ZYG11B dimerization, and the controlled heterodimer-based assays represent an important direction for future studies.

3) The manuscript discussion mentions “a valuable framework for the development of ZYG11B-based PROTACs”. I did not find obvious ways that the manuscript would aid in PROTAC design. Again, I see two identical substrate-binding pockets consistent with previous reports. Does the dimer interface create another potentially ligandable site? For KLHDC2, tetramerization might explain relatively weak PROTACs (or ligands for PROTACs) if the PROTACs/ligands are not capable of disrupting the tetramer, as the peptides do. From the current manuscript, it does not appear that ZYG11B dimerization would present a problem or advantage for PROTACs. Please elaborate if there are specific ideas.

We appreciate the reviewer’s thoughtful comment and agree that the implications of our study for PROTAC development should be articulated more carefully. We have therefore revised the Discussion to clarify this point.

While our study does not directly assess PROTAC efficacy, the high-accuracy structure of full-length ZYG11B reveals features not observed in previously reported truncated structures, including an extended surface groove that may represent a potentially ligandable site (below figure and Supplementary Fig.8).

In addition, our biochemical and cellular data indicate that ZYG11B dimerization enhances E3 ligase activity, raising the possibility that small molecules capable of modulating the monomer-dimer equilibrium of CRL2^{ZYG11B} could influence ubiquitination efficiency.

We now discuss these observations more cautiously as hypothesis-generating insights rather than established design strategies for PROTAC development (revised Discussion, lines 266-276).

4) Supplemental Figure 1 (and methods): It's not clear to me which conditions had the substrate peptide. Parts A through C of the figure? Did the authors perform experiments to determine whether the substrate peptide influences the monomer-dimer state?

We thank the reviewer for raising this point. The substrate peptide was added only immediately prior to cryo-EM grid freezing and was not present during protein purification (Supplementary Fig. 1B vs. 1A and 1C). In the revised manuscript, we have clarified this explicitly in the figure legends (lines 641-652) and in the Methods section (lines 322-324 and 329-331).

In addition, we collected cryo-EM datasets for ZYG11B-EloB-EloC complexes prepared without substrate peptide. Two-dimensional (2D) classification shows that both monomeric and dimeric states are present in the absence and presence of substrate peptide, indicating that peptide binding does not determine the oligomeric distribution (below figure).

Minor comments:

5) Methods, line 275: please provide more detail on the pCDH-puro vector. CMV promoter? There are many versions of this vector. The promoter in the plasmid matters since the level of protein overexpression in cells can have important implications.

Thank you for this suggestion. In the revised manuscript, we have specified the exact pCDH-puro construct used, which contains a CMV promoter driving ZYG11B expression. All variants were expressed from the same vector to ensure comparable transcriptional regulation (Methods, lines 294).

6) Methods, line 266: “C-terminal HIS-GFP-TEV tag” is confusing to me. It is best to list proteins in N to C order. So, was it ZYG11B-TEV-GFP-6HIS, leaving no tag behind after cleavage?

Thank you for pointing out this ambiguity. The construct is now clearly described as ZYG11B-TEV-GFP-6×His from N-terminus to C-terminus. Following TEV protease cleavage, the GFP-6×His tag is removed, leaving untagged ZYG11B (Methods, lines 285).

7) Figure 5: The proteasome cartoon would be better if made simpler (just one shape) instead of appearing as 10 stacked hexamers, which is not accurate. Or, perhaps Nature Publishing can provide a better proteasome cartoon that more accurately depicts the lid, base, and CP parts of the proteasome. Also, the color used for Ub makes it difficult to see above the background.

We thank the reviewer for this helpful suggestion. We simplified the proteasome cartoon to and adjusted the Ub coloring to improve visibility. These changes are reflected in the revised Figure 5.

Reviewer #2 (Remarks to the Author):

This manuscript presents structural and functional analyses of the human CRL2^{ZYG11B} E3 ubiquitin ligase, focusing on how ZYG11B engages substrates bearing Gly/N-degrons. Using cryo-electron microscopy, the authors report structures of full-length ZYG11B bound to the EloB-EloC adaptor complex and a Gly/N-degron peptide in both monomeric and dimeric assemblies. Complementary cell-based assays support the notion that disrupting substrate, adaptor, or dimerization interfaces impairs substrate degradation. While the study offers some new observations, there are several major concerns that should be addressed to clarify the novelty and rigor of the findings.

We sincerely thank the reviewer for the careful evaluation and thoughtful summary of our study. In response to the concerns raised, we have performed additional experiments and substantially revised the manuscript to clarify the novelty, improve the rigor of data presentation, and more appropriately frame our conclusions.

First, a closely related study was posted on bioRxiv in 2024 and is only briefly mentioned in the Discussion. The authors do not adequately acknowledge that the previously reported structures overlap significantly with those presented here. This raises concerns about the originality of the work and diminishes its novelty. The manuscript should provide a more transparent comparison to the bioRxiv study and clearly articulate what is uniquely contributed here.

We thank the reviewer for raising this important point. Following the reviewer's suggestion, we have revised the manuscript to more explicitly acknowledge the overlap with the recent bioRxiv study and to clarify the distinct contributions of our work.

The bioRxiv study reports structures of CRL2^{ZYG11B} in complex with extended degron peptides derived from NLRP1 and ORF10, revealing the mode of substrate recognition and holoenzyme assembly by monomeric CRL2^{ZYG11B}. In contrast, our study captures a previously unreported dimeric assembly of ZYG11B that persists within the context of the fully assembled CRL2^{ZYG11B} holoenzyme.

Importantly, we provide functional validation of this dimeric state using both cellular degradation assays (Fig.4h) and a fully reconstituted *in vitro* ubiquitination system (Supplementary Fig.7a, b). These data demonstrate that ZYG11B dimerization enhances E3 ligase activity, an aspect not addressed in the bioRxiv study. We now explicitly state these distinctions in the revised Discussion (lines 219-227).

Second, the reported ZYG11B dimer represents a potential novel finding not described in the bioRxiv paper. However, the biological relevance of this dimer is not convincingly established. Specifically, the manuscript lacks structural data of the full CRL2^{ZYG11B} complex including Cul2, which would be necessary to demonstrate that the dimeric form is retained upon assembly with the full E3 ligase machinery. The only supporting evidence—weak density corresponding to Cul2 in 2D class averages—is not sufficient. Stronger experimental validation, such as reconstruction of the dimeric complex with Cul2-Rbx1, is needed to substantiate this claim.

We thank the reviewer for this important comment and agree that cryo-EM alone cannot fully exclude the possibility of non-physiological assemblies. To address this concern, we therefore complemented our structural observations with independent biochemical and functional assays.

Analytical ultracentrifugation (AUC) reveals the coexistence of monomeric and dimeric CRL2^{ZYG11B}

assemblies in solution (below figure, panel-a), consistent with the populations observed in cryo-EM 2D classifications (below figure, panel-b).

Moreover, targeted disruption of the dimer interface selectively impairs substrate ubiquitination in a reconstituted *in vitro* system (below figure, panel-a) and substrate degradation in cells, without affecting complex assembly or subcellular localization (below figure, panel-b and c).

While we were unable to obtain a high-resolution reconstruction of the full dimeric CRL2^{ZYG11B} complex despite extensive cryo-EM data collection, the concordance between structural, biochemical, and cellular data supports the conclusion that the observed dimeric state represents a functionally relevant assembly rather than a cryo-EM artifact.

Third, the structural data are not presented with sufficient clarity or transparency to allow readers to evaluate the quality of the reconstructions. The reported resolutions (~4 Å) are modest and may not support unambiguous side chain assignments, yet the manuscript contains detailed interaction descriptions (e.g., hydrogen bonding networks) that may go beyond what is justified by the data.

We agree with the reviewer that side-chain level interactions should be interpreted conservatively at the originally reported resolution. In response, we reprocessed the cryo-EM datasets, improving the resolution to 3.37 Å for the monomeric state and 3.27 Å for the dimeric state (see Supplementary Table 1 and new PDB validation reports). Even with these improvements, we have deliberately limited detailed interaction descriptions to regions supported by clear and continuous density. Speculative hydrogen-bond assignments have been removed, and the interaction analysis has been carefully re-evaluated. Corresponding figures have been re-rendered to reflect these revisions (lines 135-146 and 179-189, and Figs. 2C, 3c-e, 3j, and 4e-g).

With the exception of Fig. 1, cryo-EM density maps are not shown in the main figures. Furthermore, figures such as 2C, 3a, 3c-e, 3j, and 4e-f appear to use surface renderings derived from atomic models, not the experimental maps themselves, which could mislead readers about the level of structural detail actually supported by the data.

In the original submission, overall cryo-EM density maps were shown in Figure 1, with additional regional density maps provided in Supplementary Figure 3. In the revised manuscript, these figures have been updated to display higher-resolution maps corresponding to the reprocessed datasets (Fig. 1 and Supplementary Fig. 3).

In addition, we now explicitly show experimental cryo-EM density maps for key interaction interfaces and have removed the dot-style surface representations used in the original version, which may blur the distinction between experimental data and model-derived surfaces. These revisions are intended to improve transparency and allow readers to better assess the quality of the reconstructions (blow figure associated with Figs. 3c-e, and 4e-g).

The authors state that AlphaFold models were used for initial model building, but it remains unclear to what extent the final structural conclusions rely on experimental evidence versus predicted models.

We thank the reviewer for highlighting this point. AlphaFold predictions were used solely as a reference to assist initial model building and domain assignment. All major structural conclusions are based on experimentally determined cryo-EM maps.

Notably, the experimentally derived ZYG11B structure aligns well with the AlphaFold prediction (C α RMSD of 1.56 Å); however, all mechanistic interpretations are drawn exclusively from experimental density rather than from predicted models. We have clarified this explicitly in the revised text and figure legends (lines 100-102; Supplementary Fig. 4b).

Fourth, the GPS degradation assays are insufficiently described and potentially misleading. The presentation of the data lacks normalization details; the Y-axis scale and its relation to population-level distributions are unclear.

We thank the reviewer for raising this concern. The GPS (Global Protein Stability) assay is a flow-cytometry–based dual-fluorescence reporter system in which protein stability is quantified by the GFP/DsRed ratio, with DsRed serving as an internal expression control (*Science*. 2008, 322(5903):918-923). This method was originally developed by the Stephen Elledge laboratory and has been widely applied (*Science*, 2019, 365(6448):eaaw4912; *Mol Cell*, 2021, 81(16):3262-3274.e3; *Nat Commun*. 2024, 15(1):3558).

In our experiments, cells were first gated on FSC-A and SSC-A to exclude debris and doublets. DsRed-positive cells were then defined as events with DsRed fluorescence intensity > 1×10^3 , corresponding to

cells carrying lentivirally integrated GPS reporters (Step1). GFP/DsRed fluorescence ratios were subsequently calculated in FlowJo to quantify the stability of the GFP-fused substrate (Step2). In the resulting distributions, the y-axis represents the normalized frequency of cells rather than absolute event counts. Specifically, event counts were normalized such that the total frequency sums to 100% (Step3), thereby reflecting the population-level distribution of relative substrate stability (blew figure). We have now added a detailed description of these procedures to the revised Methods section (lines 398-407).

Mutant constructs show effects beyond the vector control, which suggests potential dominant-negative behavior that is not explored or acknowledged.

We agree that dominant-negative effects—where mutant protein interferes with the function of wild-type protein—may contribute to the observed phenotypes, particularly given the presence of endogenous ZYG11B in the reporter cells and the oligomeric nature of the complex. Following the reviewer's suggestion, we have now explicitly acknowledged this possibility in the revised manuscript (lines 166-169).

The influence of variable expression levels, as seen in Western blots, is not discussed.

To address this concern, we re-established stable cell lines expressing either wild-type ZYG11B or the dimer-interface mutant and selected pools with comparable expression levels. As shown in the revised manuscript (blow figure and Fig. 4h), even when expressed at matched levels, the dimer-interface mutant consistently exhibits impaired degradation of degraon-fused substrates relative to wild-type ZYG11B.

Moreover, the choice of mutations-particularly the combination of L150S/D152S/R153S/K223A/F418A/P419A/N420A-seems arbitrary and is the only construct used to support the functional relevance of dimerization. Additional mutants or rescue experiments would be helpful to strengthen this conclusion.

We thank the reviewer for this comment. We initially tested several mutants containing fewer substitutions at the dimer interface; however, these variants did not produce measurable phenotypes in the GPS assay (below figure). Given the large buried surface area of the dimer interface (~1358 Å²), we therefore designed a combined multi-site mutant to effectively disrupt dimerization.

Importantly, this multi-site mutant does not alter subcellular localization or assembly with other CRL2 components, yet robustly impairs E3 ligase activity both *in vitro* and in cells (as discussed in our response to Comment 2). Together, these data support the functional relevance of the ZYG11B dimerization.

Minor points:

Line 10 1: The term “electron density” is not appropriate in the context of cryo-EM and should be replaced with “electrostatic potential map” or “Coulomb potential.”

We thank the reviewer for this suggestion. We removed the term “electron density” in the revised manuscript and deleted the sentence containing “...dimeric structure due to its higher resolution...”, because the resolutions of both the monomeric and dimeric structures have been improved.

The title is somewhat misleading; the study focuses ¹³ on the ZYG11B-EloB-EloC complex and substrate

recognition. The involvement of the full CRL2 ligase (including Cul2 and Rbx1) is limited and not structurally characterized in depth.

We appreciate this suggestion and have revised the title to: “Structures of ZYG11B-EloB-EloC-substrate complex reveal mechanisms of CRL2^{ZYG11B} assembly and function”.

In summary, while the manuscript contains potentially interesting observations, the novelty relative to prior work, the interpretation of the structural data, and the functional validation all require clarification or strengthening to meet the standards of a high-impact journal.

We thank the reviewer for the constructive and detailed critique. In response, we have performed additional experiments, reprocessed structural data, and substantially revised the manuscript to clarify novelty, improve transparency, and strengthen functional validation. We believe these revisions address the reviewer’s concerns and improve the rigor of the study.

We sincerely thank the reviewers for their constructive suggestions and continued support. In the revised manuscript, we have incorporated additional experimental data and clarifications to address the remaining concerns. Detailed point-by-point responses are provided below (reviewers' comments are shown in red and our responses in black). Major revisions in the manuscript are indicated as underlined text.

Point-by-Point Response:

Reviewer #1 (Remarks to the Author):

The authors nicely addressed my comments from the original review.

I will only comment further on the cell-free ubiquitination experiments (new data; Supplemental Figure 7). The legend describes the ubiquitination as “robust”. It is not robust so I suggest alternative language. Sometimes these cell-free reactions are weak but if the controls are consistent with E3-dependent ubiquitination (see below), I'd deem these studies acceptable. A couple of notes/suggestions that could enhance cell-free activity. 20 mM ATP is a high concentration—and ATP can bring down the pH of the final reaction substantially if the ATP stock is not pH adjusted or buffered enough. I'd suggest testing 2 mM ATP final and be mindful of the pH. One could also adjust the concentrations of the enzymes and substrate.

We thank the reviewer for the continued support and for the valuable suggestions to further improve the biochemical assays. We agree that the ubiquitination activity observed in our initial experiments was modest, as predominantly mono- and di-ubiquitinated substrates were detected without extensive polyubiquitination. Following the reviewer's recommendation, we optimized reaction conditions, including ATP pH and concentration as well as enzyme and substrate levels. Under the revised conditions (5 mM ATP, pH-adjusted), substrates remained to be primarily mono- and di-ubiquitinated. However, the dimer-interface mutant consistently exhibited markedly reduced activity compared with wild-type CRL2^{ZYG11B} (below figure and updated Supplementary Fig 7b). These optimizations improve reaction consistency while preserving the key comparative conclusion between wild-type and mutant complexes. We have revised the legend of

Supplementary Figure 7b (updated) to more accurately describe the data as follows: “Wild-type CRL2^{ZYG11B} promoted detectable mono- and di-ubiquitination of the substrate within 5 min, whereas the dimer-disrupting mutant showed weaker ubiquitination at all examined time points” (lines 707–710).

Also, how were the specific ubiquitin conjugating enzymes chosen? Add a short commentary and cite Li et al. 2024 (PMID: 38382526) or another reference.

We thank the reviewer for this suggestion. The E2 enzymes (UBE2R1 and UBE2D3) were selected based on previous reports demonstrating that this combination enhances CRL2-mediated ubiquitination efficiency (PMID: 38360992). We have now added a brief explanation and appropriate citations in the revised manuscript (lines 414).

In the Methods section, the authors write that “reaction mixtures without ATP were used as control”, but I do not see such data. This control, and ideally a control with excess substrate peptide to occupy that pocket on the E3, would strengthen the findings. The “no ATP” condition should also show a drastic difference on an anti-ubiquitin blot but the substrate peptide should not affect ubiquitination of non-substrate components (ubiquitination of E3 components is common in these types of reactions) but should reduce your model substrate ubiquitination. An alternative to the excess substrate peptide control would be to leave out the E3. Sometimes 1 or 2 ubiquitins can be added to substrates without an E3 in these types of reactions involving extremely high concentrations of everything. Better overall E3-dependent substrate ubiquitination would allow comparison of multiple concentrations of WT and mutant E3 to estimate how much dimerization impacts E3 activity.

We thank the reviewer for these constructive suggestions.

First, we performed substrate peptide-competition assays using the same degron peptide (GYFQRGK) as in the model substrate (GYFQRGK-GFP). Increasing concentrations of the competitive peptide substantially reduced substrate ubiquitination (below figure a and updated Supplementary Fig 7c), consistent with degron-dependent E3 activity.

Second, we conducted E3 concentration-gradient experiments, which demonstrated that substrate ubiquitination strictly depends on the presence and amount of CRL2^{ZYG11B} (below figure b and updated Supplementary Fig 7d). Increased E3 concentration enhanced substrate ubiquitination efficiency, supporting enzyme-dependent catalysis.

Together, these additional controls confirm that the observed substrate ubiquitination is CRL2^{ZYG11B}-dependent. Importantly, under all tested conditions, the dimer-interface mutant consistently displayed reduced activity relative to wild-type enzyme.

Minor comment: While “ubiquitination” and “ubiquitylation” are interchangeable terms, it is good practice to use one consistently in a single article.

We thank the reviewer for pointing this out. The term “ubiquitination” is now used consistently throughout the revised manuscript.

Reviewer #2 (Remarks to the Author):

I am satisfied with the authors' revisions, which substantially improve the rigor and clarity of the manuscript. The additional structural and biochemical are largely convincing.

We sincerely thank the reviewer for the positive assessment and for the constructive feedback that significantly improved the rigor and clarity of the manuscript.

One more point: While the rebuttal explains why a multi-site mutant was necessary to abolish the dimer interface, this should be briefly stated in the manuscript (better with supplemental data) to clarify why single or limited mutations were insufficient.

We appreciate this suggestion. We have now included a brief explanation in the main text (lines 206-208 and 217), supported by supplemental data (Supplementary Fig 8), showing that single or limited interface mutations did not produce measurable functional effects.